# A Comparative Analysis of Treatment-Related Changes in the Diagnostic Biomarker Active Metalloproteinase-8 Levels in Patients with Periodontitis

**DOI:** 10.3390/diagnostics13050903

**Published:** 2023-02-27

**Authors:** Mutlu Keskin, Juulia Rintamarttunen, Emre Gülçiçek, Ismo T. Räisänen, Shipra Gupta, Taina Tervahartiala, Tommi Pätilä, Timo Sorsa

**Affiliations:** 1Oral and Dental Health Department, Altınbaş University, Istanbul 34140, Turkey; 2Department of Oral and Maxillofacial Diseases, Head and Neck Center, University of Helsinki and Helsinki University Hospital, 00290 Helsinki, Finland; 3Fulya Oral and Dental Health Clinic, Tekirdağ 59030, Turkey; 4Oral Health Sciences Centre, Post Graduate Institute of Medical Education & Research (PGIMER), Chandigarh 160012, India; 5Department of Pediatric Surgery, Children’s Hospital University, University of Helsinki and Helsinki University Hospital, 00290 Helsinki, Finland; 6Department of Oral Diseases, Karolinska Institutet, 171 77 Stockholm, Sweden

**Keywords:** periodontal diseases, diagnosis, matrix metalloproteinase(s), periodontitis, inflammation

## Abstract

Background: Previous studies have revealed the potential diagnostic utility of aMMP-8, an active form of MMP-8, in periodontal and peri-implant diseases. While non-invasive point-of-care (PoC) chairside aMMP-8 tests have shown promise in this regard, there is a dearth of literature on the evaluation of treatment response using these tests. The present study aimed to investigate treatment-related changes in aMMP-8 levels in individuals with Stage III/IV—Grade C periodontitis compared to a healthy control group, using a quantitative chairside PoC aMMP-8 test, and to determine its correlation with clinical parameters. Methods: The study included 27 adult patients (13 smoker, 14 non-smoker) with stage III/IV-grade C periodontitis and 25 healthy adult subjects. Clinical periodontal measurements, real-time PoC aMMP-8, IFMA aMMP-8, and Western immunoblot analyses were performed before and 1 month after anti-infective scaling and root planing periodontal treatment. Time 0 measurements were taken from the healthy control group to test the consistency of the diagnostic test. Results: Both PoC aMMP-8 and IFMA aMMP-8 tests showed a statistically significant decrease in aMMP-8 levels and improvement in periodontal clinical parameters following treatment (*p* < 0.05). The PoC aMMP-8 test had high diagnostic sensitivity (85.2%) and specificity (100.0%) for periodontitis and was not affected by smoking (*p* > 0.05). Treatment also reduced MMP-8 immunoreactivity and activation as demonstrated by Western immunoblot analysis. Conclusion: The PoC aMMP-8 test shows promise as a useful tool for the real-time diagnosis and monitoring of periodontal therapy.

## 1. Introduction

Periodontitis is a chronic inflammatory disease that affects the tissues that support the teeth and is extremely prevalent in the community [1,2]. The pathogenic evolution of the dysbiotic microbial structure in the dental biofilm is among the most crucial factors in the onset and progression of periodontal disease. This process then leads to the continuation of tissue destruction as a result of the host response’s non-physiologic overreaction [2]. Periodontitis, one of the most common causes of tooth loss, is not only limited to local tissues but has been linked to a variety of systemic diseases, including diabetes, cardiovascular disease, cancer, and Alzheimer’s disease [3,4,5,6,7,8]. As a result, early diagnosis of the inflammatory periodontal disease process is critical in preventing tissue destruction [9,10].

Matrix metalloproteinases (MMPs), a family of genetically distinct but structurally related proteases that can degrade almost all extracellular matrix (ECM) structures, play an important role in tissue destruction caused by degenerative periodontal diseases [11]. MMPs can also process non-matrix bioactive molecules affecting immune responses [12]. These non-matrix bioactive molecules include, but are not limited to serpins, pro- and anti-inflammatory cytokines and chemokines, growth factors, complement components and insulin-receptor, and thereby MMPs can modify immune responses and systemic diseases [10,12]. Currently, 23 MMPs have been found to be expressed (released) in humans. MMP-8, also known as collagenase-2, is a pro-enzyme that is primarily derived from neutrophils [10]. It can be activated by microbial virulence factors, proinflammatory cytokines, and reactive oxygen species. Numerous studies have focused on MMP-8 as a diagnostic biomarker for periodontal diseases, and it has been found in oral fluids, such as mouth rinse, saliva, gingival crevicular fluid (GCF), and peri-implantitis sulcular fluid (PISF) [10]. MMP-8 levels in these fluids have been shown to correlate with the severity of periodontal and peri-implant diseases [10,13,14,15,16,17].

MMP-8 is produced and expressed during the neutrophils’ development and maturation in the bone marrow and is stored in subcellular neutrophilic granules in a latent state. When infection-induced inflammatory periodontal and peri-implant diseases appear, the process of selective degranulation and extracellular proMMP-8 release and activation begins [12,18,19]. MMP-8 has been found to be the most common collagenolytic protease in the diseased periodontium and peri-implantium [10,20,21,22,23]. 

The active form of MMP-8 which is called the active MMP-8, or aMMP-8, is the main mediator of the active tissue destruction process in inflammatory periodontal and peri-implant diseases [10,22]. The aMMP-8 levels in intraoral fluids (mouth rinse, saliva, i.e., GCF and PISF) have been found to rise in inflammatory periodontal and peri-implant diseases, i.e., [23,24,25]; aMMP-8 is regarded to be among the key biomarkers that play an important role in the diagnosis of periodontal and peri-implant diseases and has been implemented as a biomarker into the new classification of these diseases [10,12,17,21,26].

Traditional methods for diagnosing periodontal diseases include bleeding on probing, clinical attachment level measurement, probing depth, and radiographic findings [2,27]. Classical periodontal examination methods can be painful for the patient, and they are time-consuming procedures that must be repeated in all follow-up processes after periodontal treatment, which adds to bacteremia [2]. Furthermore, probing related evaluations such as bleeding on probing, pocket depth, and so on may not yield objective results due to a variety of factors, such as the force applied by the examiner and the characteristics of the periodontal probe, etc. Hence, the classical clinical assessments have been regarded to be at least partially erroneous [2,18,27,28]. On the other hand, radiographic examination methods can only provide information about the destructive effects of periodontal disease which have occurred in the past [2,29].

When considered a stand-alone evaluation criterion, bleeding on probing (BOP) values, which are considered as the gold standard for assessing periodontal disease activity, may thus be ineffective in diagnosing active periodontitis [30]. A number of longitudinal studies have also shown that BOP alone is not a good predictor of periodontal tissue destruction in treated cases [31,32]. Several studies have shown that a chairside PoC aMMP-8 test could be more effective in the diagnosis of subclinical periodontal diseases compared to BOP [33,34,35,36,37].

There are a few studies characterizing the periodontal treatment-related changes of aMMP-8, which give promising results about periodontal disease activity and evaluating its correlation with other oral biomarkers in the literature [23,24,38]. 

The present study aimed to investigate treatment-related changes in aMMP-8 levels in individuals with periodontitis using quantitative a chairside PoC aMMP-8 test and its on-line and real-time quantitative correlation with the studied clinical periodontal parameters. Consistency characteristics of the diagnostic tests were evaluated. The aMMP-8 levels and molecular forms were also assessed by IFMA and Western immunoblotting analysis, respectively.

## 2. Materials and Methods

### 2.1. Study Population and Design

The study design is presented in Figure 1. A total of 27 patients visiting a private clinic “Özel Fulya Ağız ve Diş Sağlığı Kliniği” in Tekirdağ, Turkey for their periodontal problems were recruited in the present study. The study was approved by the Biruni University Ethics Committee (2015-KAEK-71-22-06) and was carried out according to the principles of the Declaration of Helsinki. Oral and written consent was obtained from all recruited subjects. The inclusion criteria for the study were: interdental clinical attachment loss: ≥5 mm (at the site of greatest loss), detection of radiographic bone loss extending beyond 33% of the root, tooth loss due to periodontitis: ≤4 teeth (Stage III Periodontitis), ≥5 teeth (Stage IV Periodontitis). Patients with Acquired Immune Deficiency Syndrome (AIDS), uncontrolled diabetes (HbA1c > 7), and other immune-system-related chronic diseases (Crohn’s disease, etc.) were excluded from the study. Pregnant or lactating females and individuals who had received periodontal treatment within the last year were also excluded. A total of 25 systemically and periodontally healthy dental students from the University of Helsinki, Finland served as healthy controls.

### 2.2. Periodontal Examination Procedure

Comprehensive periodontal examination was performed at baseline and 1 month following periodontal treatment by a single periodontist (M.K.). Probing depths (PD) were measured at six sites of each tooth with a Williams color-coded Michigan probe. Plaque index was recorded by assigning a score of 0–3 to each surface, and average oral plaque score was calculated for each patient [39]. The percentage of bleeding on probing (BoP) was determined after probing depth measurements. Gingival margin levels (GML) were determined by taking the enamel–cement junction (ECJ) into account during probing depth measurements. The areas where the free gingival margin ended at the apical of the EJC were recorded as positive values, and the areas where the free gingival margin terminated at the coronal point were recorded as negative values. Clinical attachment levels for each site were determined as the sum of GML and PD.

### 2.3. Periodontal Treatment Procedure

Periodontal treatment was carried out by a specialist periodontist (M.K.). Initially, cause-related therapy, including full-mouth scaling and root planing procedures, were performed along with oral hygiene instructions. At 2 weeks following the non-surgical phase of the periodontal therapy, periodontal sites associated with irregular bony contours, angular defects, or pockets in which a complete access with non-surgical periodontal therapy was not possible, such as grade II–III furcation defects, were treated with open flap debridement. Patients who underwent the surgical phase of treatment were prescribed amoxicillin plus clavulanic acid (1gr/day) and chlorhexidine mouth rinse (0.12%) twice a day for 7 days and recalled thereafter for suture removal. All patients were re-evaluated clinically 1 month following treatment.

### 2.4. Quantitative Chairside PoC aMMP-8 Analyses

Levels of aMMP-8 were measured quantitatively using rapid PoC chairside aMMP-8 kits (Periosafe^®^, Dentognostics GmbH, Solingen, Germany) and a quantitative spectrometer analyzer (Oralyzer^®^, Dentognostics GmbH, Solingen, Germany) on mouth rinse samples collected before treatment and 1 month following periodontal treatment. To perform a comparative analysis with the periodontitis patient group, analysis of aMMP-8 was also conducted on the healthy control group at T0 (baseline). PoC chairside aMMP-8 analyses were performed prior to clinical measurements, and manufacturer’s instructions were followed. It was recommended that patients and controls not eat for 1 h before analyses. First, the patients and controls were instructed to rinse their mouths with clean water (drinking or distilled water) for 30 s and spit it out. After a waiting period of 1 min, they were told to rinse their mouths for 30 s with 5 mL of distilled water in the aMMP-8 kit (Periosafe^®^) and spit it back into the container. Then, 3–4 drops were taken from the container with a sterile syringe and poured into the well on the test cassette provided in the aMMP-8 kit. Immediately after that, the cassette was transferred to the digital spectrometer device (Oralyzer^®^) and quantitative results were obtained after 5 min. The remaining liquid in the container was transferred to Eppendorf tubes and stored at −70 °C for further laboratory analysis.

### 2.5. Measurement of the aMMP-8 Levels by Immunofluorometric Assay (IFMA)

The aMMP-8 level from mouth rinse samples was determined by a time-resolved immunofluorescence assay (IFMA) as described by Öztürk et al. [40]. Briefly, aMMP-8-specific monoclonal antibodies 8708 and 8706 (Actim Oy, Espoo, Finland) were used in the analysis as a catching antibody and a tracer antibody, respectively. In this protocol, the diluted samples were allowed to incubate for 1 h with the Europium labelled tracer antibody. The fluorescence was measured using an EnVision 2015 multimode reader (PerkinElmer, Turku, Finland). 

### 2.6. Western-Immunoblotting Testing Procedure

The molecular forms of MMP-8 were detected from mouth rinse samples by a modified enhanced chemiluminescence (ECL) Western blotting kit according to protocols recommended by the manufacturer (GE Healthcare, Amersham, UK) as described earlier by Rautava et al. [41]. Briefly, the proteins of mouth rinse samples were first separated by electrophoresis and then electro-transferred onto nitrocellulose membranes Protran (Whatman GmbH, Dassel, Germany). The membranes were incubated overnight with monoclonal primary antibodies anti-MMP-8 [42] and then with horseradish peroxidase-linked secondary antibody (GE Healthcare, Buckinghamshire, UK) for 1 h. The membranes were washed 4 times in TBST between each step for 15 min. The proteins were visualized using the ECL system according to protocol. The recombinant human MMP-8 (100 ng, Calbiochem, Darnstadt, Germany) was used as a positive control.

### 2.7. Statistical Analysis

All periodontal parameters, including probing depth, bleeding on probing, plaque index, and clinical attachment level were examined before periodontal treatment and 1 month following anti-infective periodontal treatment. Normality tests were performed to test the normality of the data before calculating paired samples t-tests. Paired-samples *t*-test was used to analyze the statistically significant differences between these two phases. The effect of smoking on aMMP-8 levels was tested with repeated measures ANOVA. A *p* < 0.05 was accepted as statistically significant value.

Receiver operating characteristic (ROC) analysis and the area under the ROC curve (AUC) were used to examine the diagnostic accuracy of aMMP-8 to classify periodontitis and periodontally healthy subjects. In order to identify optimal cut-offs from the ROC curves, the Youden Index was used for calculating diagnostic sensitivity and specificity (Se and Sp).

## 3. Results

### 3.1. Study Population

A total of 27 periodontitis patients (4 = Stage III, 23 = Stage IV, 27 = Grade C) and 25 healthy control subjects were enrolled in the study. Ages of periodontitis patients ranged between 30 and 70 years. All healthy subjects were younger (age range 23 to 25 years) than the study group (*p* < 0.01). Demographic characteristics of periodontitis patients and healthy control subjects are shown in Table 1.

### 3.2. Clinical Periodontal Parameters

All periodontitis patients were subjected to non-surgical periodontal therapy followed by open flap debridement in seven of them. Statistically significant improvements following anti-infective treatment were observed for all periodontal parameters (*p* < 0.001) (Table 2). Scatter plot diagrams of the relationship between probing depths, bleeding on probing, clinical attachment level, and plaque indices before periodontal treatment and after anti-infective periodontal treatment are presented in Figure 2. The clinical parameters as well as aMMP-8 levels of the periodontitis patients reduced to levels close to that of healthy subjects following periodontal therapy (Table 2). 

Both non-smoker and smoker subjects showed statistically significant decreases in terms of inflammatory clinical parameters (*p* < 0.001). A similar clinical healing pattern was observed in both groups (Figure 3).

### 3.3. aMMP-8 Results

A statistically significant decrease in oral rinse aMMP-8 levels following anti-infective periodontal treatment was observed regarding both Oralyzer^®^ and IFMA results and in correlation with bleeding on probing (*p* < 0.05) (Table 2 and Figure 4). Both Oralyzer^®^ and IFMA results indicated a similar pattern of decrease in terms of oral rinse aMMP-8 levels, and it was also observed that smoking did not have a significant effect on aMMP-8 PoC testing (Figure 4 and Figure 5) (*p* > 0.05).

An ROC analysis was used for analyzing the diagnostic ability of aMMP-8 PoC and IFMA tests to discriminate patients with periodontitis (before treatment) from healthy controls (Figure 6).

AUC was also calculated and showed excellent discrimination ability between periodontitis and periodontally healthy groups (aMMP-8 POC test = 0.963; 95% CI: 0.904–1.000; *p* < 0.001 and aMMP-8 IFMA test = 0.975; 95% CI: 0.941–1.000; *p* < 0.001). Optimal cut-offs for aMMP-8 POC and IFMA tests were estimated by Youden’s Index (aMMP-8 POC test: 20.0 ng/mL; sensitivity: 0.852; specificity: 1.000; aMMP-8 IFMA test: 43.20 ng/mL; sensitivity: 0.926; specificity: 0.920).

With the cut-off set at 20 ng/mL, pretreatment sensitivity was 85.2% and post-treatment sensitivity was 81.5%; 85.2% (23 out of 27) of study subjects were aMMP-8 positives (>20 ng/mL), and 78.3% (18 out of 23) of aMMP-8 positive patients were converted to aMMP-8 negatives (<20 ng/mL) following periodontal therapy.

With the cut-off set at 10 ng/mL, pretreatment sensitivity was 100%. All (27 out of 27) study subjects were aMMP-8 positives (>10 ng/mL), and 43.4% (10 out of 23) aMMP-8 positive subjects converted to aMMP-8 negatives (<10 ng/mL) following therapy(Table 3).

### 3.4. Western Immunoblotting Analysis Results

Representative Western immunoblot analysis and aMMP-8 POC-test outcomes of MMP-8 in the studied mouth rinse samples from orally and systemically healthy and diseased study subjects are shown in Figure 7. MMP-8 was in latent form in the healthy sample (Figure 7A, Lane 2), and in the diseased samples it was converted to active and fragmented forms (Lane 3) as analyzed by monoclonal anti-MMP-8 antibody (Figure 7A). Negative (−, <20 ng/mL) and positive (+, ≥20 ng/mL) aMMP-8 POC-test outcomes are shown in Figure 7B.

## 4. Discussion

Periodontal diseases are chronic inflammatory conditions affecting the supporting tissues of the teeth [1]. One of the key enzymes involved in the breakdown of these tissues is matrix metalloproteinase-8 (MMP-8). While MMP-8 is important for normal tissue remodeling and repair, excessive or uncontrolled production of this enzyme can lead to tissue destruction and the progression of periodontal diseases. Recent studies have focused on the use of aMMP-8 as a biomarker for periodontal diseases; aMMP-8 refers to the active form of MMP-8, which is produced by neutrophils and other inflammatory cells in response to bacterial infection. Elevated levels of aMMP-8 have been linked to increased tissue destruction and disease progression in periodontal diseases, making it a valuable diagnostic and prognostic tool for these conditions [12]. Furthermore, aMMP-8 has been shown to be a more specific marker for active periodontal disease than total MMP-8, which can be found in both active and inactive forms [12,21].

The present study aimed to evaluate treatment-related changes of mouth rinse aMMP-8 levels by using PoC aMMP-8 kits and Oralyser-reader, which is a non-invasive method that rapidly and quantitatively produces chairside on-line and real-time results. Both chairside PoC aMMP-8 tests and IFMA aMMP-8 laboratory analysis confirmed that pre-treatment mouth rinse aMMP-8 levels were clearly higher than mouth rinse levels of patients after 1 month following periodontal treatment. 

Our study provides valuable insights into the potential use of PoC chairside aMMP-8 tests and IFMA aMMP-8 laboratory analysis in the diagnosis and post-treatment follow-up of periodontal diseases. However, there are several limitations that should be considered when interpreting the results. Firstly, the small sample size could limit the generalizability of our findings. Secondly, the short follow-up period of only 1 month limits the assessment of the effectiveness of these techniques over time. On the other hand, the absence of periodontally healthy smokers in our study groups can be considered a limitation in comparative evaluations. Despite these limitations, our study provides important insights into the potential use of PoC chairside aMMP-8 tests and IFMA aMMP-8 laboratory analysis in the diagnosis and post-treatment follow-up of periodontal diseases.

This study utilized both the aMMP-8 PoC chairside aMMP-8 test and the aMMP-8 IFMA measurements that utilize the same monoclonal antibodies (Sorsa T et al., US patent no: US10488415B2). These techniques utilize two monoclonal, i.e., primary or catching antibody and secondary or detection [9,17,23,43,44]. Despite that, they correlate with each other; the techniques produced different values evidencing that both techniques can independently diagnose and differentiate periodontal health and disease. Both techniques can also be applied to monitor the treatment of the disease [12,24,35]. This study thus confirms and further extends the results of several previous studies demonstrating the potential benefits of POC chairside aMMP-8 and IFMA aMMP-8 laboratory analysis in terms of diagnostic distinction between periodontal health and disease [34,36,37,40,45,46,47,48]. Furthermore, our present findings are in accordance with numerous studies linking elevated oral aMMP-8, but not total MMP-8, to active and progressive stages of periodontal and peri-implant diseases [20,23,43,49,50,51,52,53,54,55].

It was previously shown that smokers had significantly higher levels of aMMP-8 in their saliva compared to ex-smokers or non-smokers [17,54]. When the pre-periodontal treatment results were evaluated from a diagnostic point of view, smoking was not found to significantly affect the aMMP-8 PoC testing being in agreement with previous studies on aMMP-8 in oral fluids (Mäntylä et al., 2006). The sensitivity of the test was found to be 85.2% when the cut-off value was determined to be 20 ng/mL. According to a recently published study of Öztürk VÖ et al. [40], in which they included Stage III and IV periodontitis patients, diagnostic sensitivity of PoC aMMP-8 was observed as 83.9% [40]. In other studies in which periodontitis and peri-implantitis patients were included and the cut-off value was determined to be 20 ng/mL, it was observed that the aMMP-8 PoC test’s sensitivity ranged between 76–90% [21]. 

Clinical periodontal parameters of pre-treatment and 1 month following periodontal treatment revealed statistically significant improvement as predicted and consistent with the literature. [56,57]. The quantitative chairside PoC aMMP-8 and IFMA aMMP-8 laboratory results both demonstrated a statistically significant decrease, correlating with and reflecting well with the clinical findings. There are many studies in the literature reporting a decrease in aMMP-8 levels following periodontal treatment [10,12,24,25,48,49,58]. While MMP-8 in its latent form was detected more frequently in the healthy state [53], the release of degranulated aMMP-8, its activated form, increases with periodontal and peri-implant inflammation and disease severity [12,23,54,55]. The statistical decrease in aMMP-8 levels post-periodontal treatment suggests that active tissue destruction, along with clinical disease activity, is reduced, confirming the role of MMP-8 in periodontitis pathogenesis [10,12,59].

When analyzing the clinical results, it becomes clear that factors, such as deep periodontal pockets, bleeding on probing (BOP), and oral hygiene, are strongly linked. However, despite treatment, not all patients were able to achieve complete oral health status as these parameters did not return to normal levels in all cases. Furthermore, it was observed that the post-treatment mouth rinse aMMP-8 levels (in both IFMA and PoC chairside aMMP-8 Tests) were higher than health-associated levels. In the study of Umeizudike et al., it was found that in the sixth month post-periodontal treatment, the aMMP-8 levels did not reach close to that of the healthy control group [48]. Literature data further suggests that individuals with gingivitis may have elevated aMMP-8 IFMA levels and aMMP-8 release may persist in the periodontal sites that respond poorly to treatment [25,35,36]. Periodontally and systemically healthy dental students without any periodontal disease experience and activity all had negative ([-], <20 ng/mL) aMMP-8 levels. This finding was also compatible with the literature which further affirms that Periosafe PoC aMMP-8 test negativity can be regarded as a biomarker of periodontal and peri-implant health [40]. Since this study includes a 1-month follow-up, the clinical and biochemical findings might not have reached the level of complete health due to persistent gingival inflammation and residual periodontal pockets. There are studies in the literature that suggest that the post-treatment re-evaluation period ranges from 2 weeks to 6 months (59, 60). Morrison et al. state that the severity of periodontitis can be significantly reduced during the 1-month post-periodontal treatment follow-up process. However, the oral hygiene process must be fully ensured to determine the ongoing treatment need [60]. 

When comparing aMMP-8 cut-off 20 ng/mL vs. 10 ng/mL [34,36], we found that especially after treatment, the periodontal health targeting was reduced with a 10 ng/mL cut-off. Deng et al. [36] used 10 ng/mL as the diagnostic cut-off value, but it should be remembered that it is not recommended by the manufacturers [21,35]. Our present results provide further support for the use of 20 ng/mL as the diagnostic cut-off value for aMMP-8 PoC tests [23,24,34,35].

Laboratory analysis of immunological inflammatory factors is considered to be the gold standard [61,62,63]. The results of PoC chairside aMMP-8 tests were consistent with those of IFMA aMMP-8 analyses, indicating that non-invasive PoC aMMP-8 analysis [12,21,34,45,64] can make a potential contribution regarding the diagnosis and periodontal therapy follow-up. However, there is a need for more longitudinal studies on the functionality of PoC chairside aMMP-8 analyses in periodontal treatment and its follow-up.

## 5. Conclusions

Observation of alarmingly high mouth rinse aMMP-8 levels in individuals with periodontitis through both point-of-care aMMP-8 and IFMA aMMP-8 analyses, and their significant decrease after anti-infective periodontal treatment, highlights the practical utility of the point-of-care aMMP-8 test for real-time diagnosis and monitoring of periodontal treatment progress.

## 6. Patents

TS is the inventor of U.S. patents 1,274,416, 5,652,223, 5,736,341, 5,864,632, 6,143,476 and US 2017/0023571A1 (issued 6 June 2019), WO 2018/060553 A1 (issued 31 May 2018), 10,488,415 B2, and US 2017/0023671A1, Japanese Patent 2016-554676 and South Korean Patent No. 10-2016-7025378.

## Figures and Tables

**Figure 1 diagnostics-13-00903-f001:**
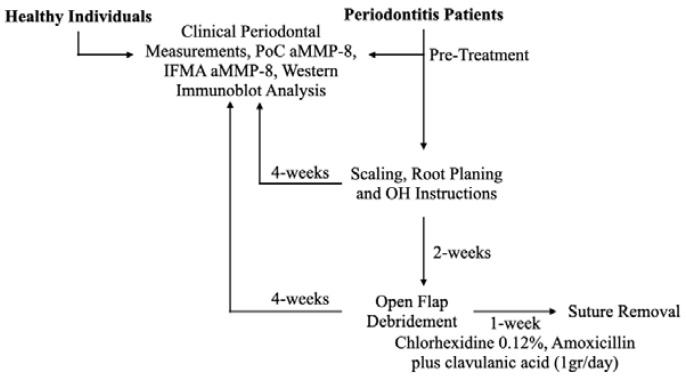
Study design.

**Figure 2 diagnostics-13-00903-f002:**
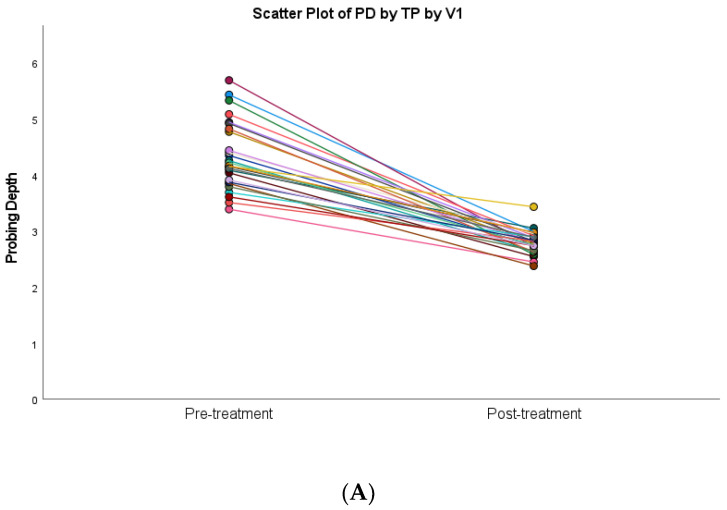
Differences in periodontal clinical parameters at baseline and 1 month after anti-infective periodontal treatment. The linear regression line is shown in different colors for each subject: (**A**) PD = probing depth; (**B**) BOP = bleeding on probing; (**C**) CAL = clinical attachment level; (**D**) PI = plaque index.

**Figure 3 diagnostics-13-00903-f003:**
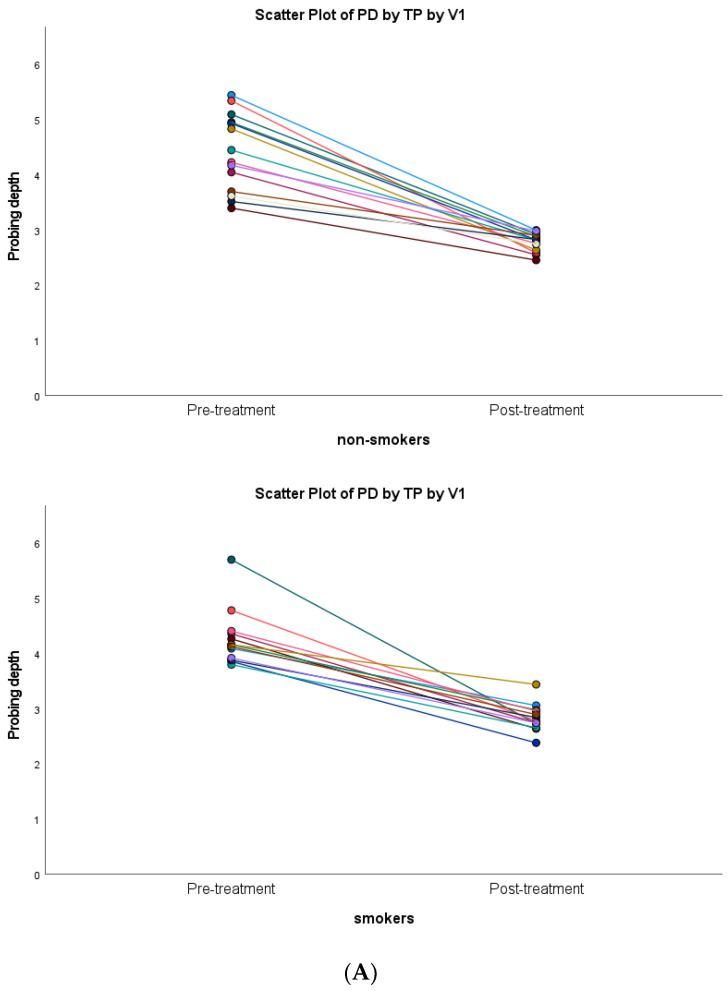
Scatter plots of the association of anti-infective periodontal treatment with smoking versus non-smoking subjects: (**A**) PD = probing depth; (**B**) BOP = bleeding on probing; (**C**) CAL = clinical attachment level; (**D**) PI = plaque index.

**Figure 4 diagnostics-13-00903-f004:**
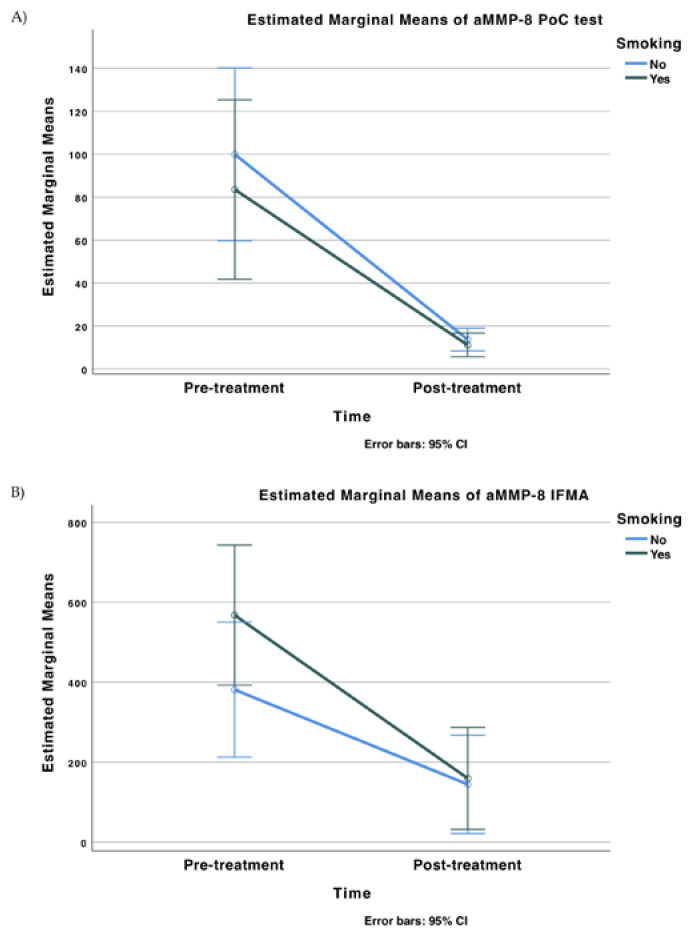
Differences in the mean levels of diagnostic marker aMMP-8 (Oralyzer^®^), IFMA aMMP-8: Pretreatment—baseline; Post-treatment—1 month following anti-infective periodontal treatment. (**A**) Estimated Marginal Means of aMMP-8 PoC test; (**B**) Estimated Marginal Means of aMMP-8 IFMA.

**Figure 5 diagnostics-13-00903-f005:**
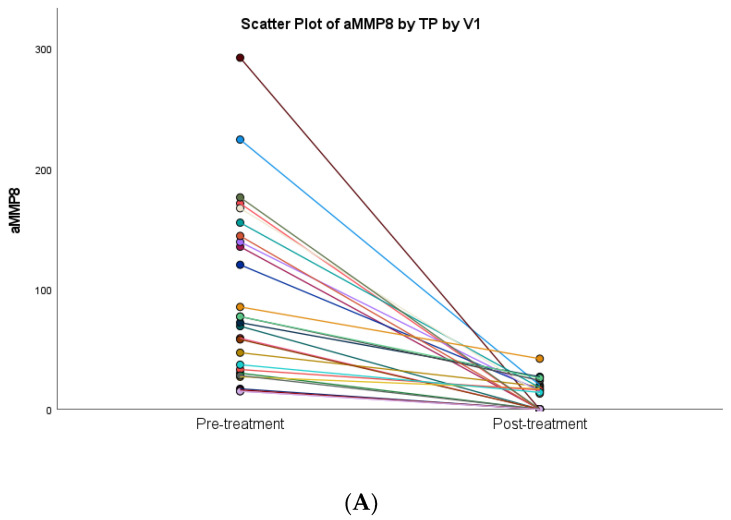
Scatter plot diagrams showing the effect of anti-infective periodontal treatment on aMMP-8 levels: (**A**) Oralyzer^®^’ (**B**) IFMA; (**C**) regression lines of means.

**Figure 6 diagnostics-13-00903-f006:**
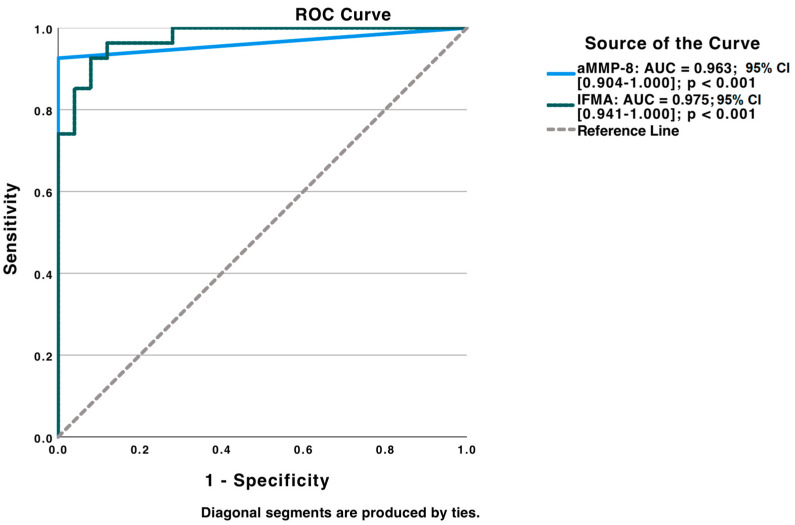
Receiver operating characteristic (ROC) analysis tested for screening diagnostic ability of aMMP-8 PoC and IFMA tests to discriminate between periodontitis and periodontal health.

**Figure 7 diagnostics-13-00903-f007:**
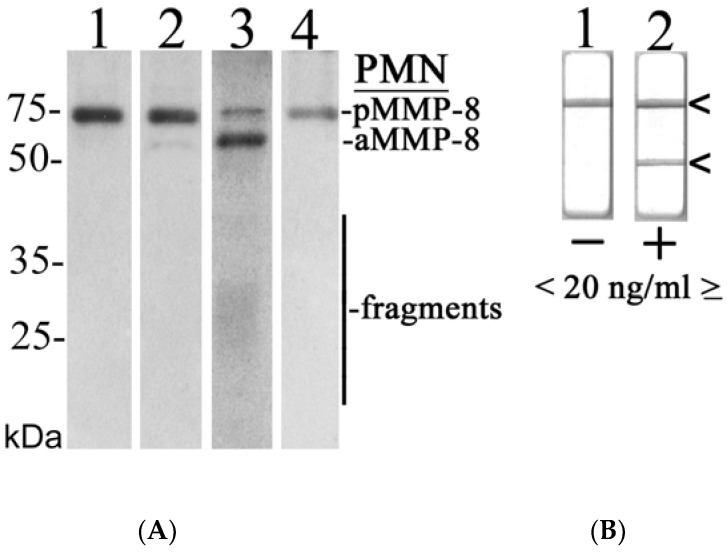
Representative Western immunoblot for molecular forms and species of MMP-8/collagenase-2 in the studied mouth rinse samples: (**A**) Lane 1: recombinant human MMP-8 (100 ng), monoclonal antibody; Lane 2: mouth rinse sample of systemically and orally healthy subject, monoclonal antibody; Lane 3: mouth rinse sample of systemically and orally/periodontally diseased subject before anti-infective treatment, monoclonal antibody; Lane 4: mouth rinse sample of systemically and orally diseased subject after anti-infective treatment; PMN indicates polymorphonuclear leukocyte; pMMP-8 indicates latent proMMP-8; aMMP-8 indicates active MMP-8; fragments indicate lower (<50 kDa) molecular size MMP-8 species due the activation and related fragmentation; (**B**) negative (−, one line, <20 ng/mL aMMP-8, Lane 1) and positive (+, two lines, ≥20 ng/mL aMMP-8, Lane 2) chairside (PoC) lateral-flow immunotest outcomes indicated by arrows on the right.

**Table 1 diagnostics-13-00903-t001:** General characteristics and periodontal classifications of study groups.

Patient Characteristics		Periodontitis	Healthy
**Age (in years)**	Mean ± SD	47.3 ± 11.84	24 ± 1.7
**Gender**			
	Females	19 (70%)	16
	Males	8 (30%)	9
**Systemic Status n (%)**			
	Healthy	22 (82%)	25 (100%)
	Cardiovascular Diseases	2 (7%)	
	Hypothyroidism	3 (11%)	
**Medication n (%)**			
	No Medication	9 (45%)	25 (100%)
	Levoythroxine Sodium	3 (15%)	
	Beta-1 selective blocker	2 (7%)	
	Atorvastatin	1 (4%)	
**Smoking (≥10 cigarettes a day, more than 5 years)**			
	Yes	13 (48%)	0 (0%)
	No	14 (52%)	25 (100%)
**Periodontitis Stage**			
	Healthy		25 (100%)
	Stage III	4 (17%)	0 (0%)
	Stage IV	23 (83%)	0 (0%)
**Periodontitis Grade**	A	0 (0%)	0 (0%)
	B	0 (0%)	0 (0%)
	C	27 (100%)	0 (0%)

**Table 2 diagnostics-13-00903-t002:** Periodontal status and active matrix metalloproteinase-8 (aMMP-8) levels for the study population of 27 periodontitis patients before and after periodontal anti-infective treatment and 25 healthy controls at baseline.

	Pre-Treatment	Post-Treatment	Significance of Comparison
BOP% *Mean (SD)*	76.46 (13.28)	5.38 (6.20)	*p* < 0.001
PD *Mean (SD)*	4.33 (0.60)	2.80 (0.21)	*p* < 0.001
PI *Mean (SD)*	1.66 (0.45)	0.86 (0.21)	*p* < 0.001
CAL *Mean (SD)*	6.51 (1.90)	5.32 (1.64)	*p* < 0.001
aMMP-8 ng/mL *Mean (SD)*	92.04 (72.19)	10.0 (11.74)	*p* < 0.001
aMMP-8 > 20 ng/mL *n (%)*	23 (85.2%)	5 (18.5%)	

**Table 3 diagnostics-13-00903-t003:** Cut-off calculated by Youden’s Index.

		Pre-Treatment		Post-Treatment	
		Count	n%	Count	n%
**PoC Chairside aMMP-8 Test** **(Cut-off [10 ng/mL])**	**Negative**	0	0%	14	51.9%
	**Positive**	27	100%	13	48.1%
**PoC Chairside aMMP-8 Test** **(Cut-off [20 ng/mL])**	**Negative**	4	14.8%	22	81.5%
	**Positive**	23	85.2%	5	18.5%

## Data Availability

Not applicable.

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
