# Peer review of "A Comparative Analysis of Treatment-Related Changes in the Diagnostic Biomarker Active Metalloproteinase-8 Levels in Patients with Periodontitis"

_diagnostics, 2023, doi:10.3390/diagnostics13050903_

Round 1

Reviewer 1 Report

The study is well conducted and the paper is deemed fit for publication in Diagnostics. 

I suggest the quality of the figures to be improved. A better resolution to the graphs provided would suffice. 

Author Response

Response to Reviewer 1:

Dear reviewer,

Thank you very much for your kind comments. The figures has been redesigned.

Reviewer 2 Report

Dear Authors,

Thank you very much for submitting you manuscript to the prestigious journal Diagnostics.

I hope that my pieces of advice and remarks will be useful in order to increase its’ scientific quality.

1.     Line 104 - Why did you include only 27 patients? Is it a premliminary study? Please motivate the low number of the sample population.

2.     Line 147 - Please rephrase the line as it may be misleading. Are you refering to the salivary levels of MMP-8 or to aMMP-8?

3.     Please increase the sizes of all images included in Fig.2, Fig. 3, Fig.4 and Fig. 5 and redesign them in the manuscript.

4.     Line 303 – Discussions section should be elaborated.

5.     Line 382 – The Conclusion section should be rephrased in such a way that it will reflect the clinical/practical impact of your study in improving current tx protocols.

Best regards!

Author Response

Dear reviewer,

We appreciate your thoughtful review of our manuscript. We have carefully considered your comments and suggestions, and we have made revisions accordingly. Please find our responses to your specific points below:

  1. Line 104 - Why did you include only 27 patients? Is it a premliminary study? Please motivate the low number of the sample population.

The role of active metalloproteinase-8 in active destructive periodontal diseases has been demonstrated in many studies. In this study, chairside POC aMMP-8 kits, which can be applied directly to the patient and provide immediate results, were used. Therefore, real-time data accumulation was provided for our patients both during the initial and treatment follow-up periods, and statistically significant differences were observed even in relatively small number of patient groups. Based on the data we obtained, in terms of the statistically significant changes in the clinical parameters and in POC (chairside) aMMP-8 observations during this follow-up period, we aimed to conduct a comparative analysis with the laboratory (IFMA) aMMP-8, which can be considered as the gold standard. On the other hand, individuals with Stage 3 and Stage 4 periodontitis can be considered as serious periodontitis, were included in this study, and patients with moderate/mild periodontitis were excluded. Systemically healthy dental students with a very healthy periodontal condition participated as the control group. Considering the literature and the clear differences in periodontal condition among our study groups, the expected difference in both the oral concentration of aMMP-8, one of the most important biomarkers of active periodontitis, and the periodontal clinical parameters before and after treatment formed the basis of our hypothesis.

  1. Line 147 - Please rephrase the line as it may be misleading. Are you refering to the salivary levels of MMP-8 or to aMMP-8?

The related sentence has been revised to:

The levels of aMMP-8 were measured quantitatively using rapid PoC chairside aMMP-8 kits (Periosafe®) and quantitative spectrometer analyzer (OraLyzer®) on mouthrinse samples collected before treatment and 1-month following periodontal treatment.

  1. Please increase the sizes of all images included in Fig.2, Fig. 3, Fig.4 and Fig. 5 and redesign them in the manuscript.

Images have been redesigned.

  1. Line 303 – Discussions section should be elaborated.

Discussion section has been revised.

  1. Line 382 – The Conclusion section should be rephrased in such a way that it will reflect the clinical/practical impact of your study in improving current tx protocols.

Conclusion section has been revised.

Reviewer 3 Report

Congrats.

Author Response

Dear reviewer,

Thank you very much for your kind comments.

Reviewer 4 Report

This paper aims to evaluate treatment-related changes of aMMP-8 levels in patients with periodontitis, testing as a diagnostic biomarker. Chairside and laboratory tests of aMMP-8 confirmed that pre-treatment levels were clearly higher than levels after 1-month following periodontal treatment.

It is a very well-structured, complete, and organized document, as well as an interesting and easy-to-read paper.

Either way, I may have some comments in the various sections:

The title should use only the non-abbreviated form of words.

The abstract should include the meaning of the abbreviated forms of words when they are first introduced in the text (e.g. MMP, IFMA, etc). The objectives are not related to the group control and the results do not mention at all. Time of measuring in this group should be explained (only time 0  measurements) and the meaning of including this group (to test the consistency of diagnostic test? Conclusions: cannot give this conclusion just with the present study results.

Introduction- phrase initiating in line 56 is too long and confusing. Please divide into shorter sentences.

Objectives should include consistency characteristics of the diagnostic tests.

Material and Methods- Quantitative chairside PoC aMMP-8 analyzes were done in control patients at what end point? Please include the purpose of introducing control group

Surgical treatment was performed in how many periodontal patients? Tests were performed 1 month after surgical treatment. Is this treatment could influence the results? Please explain better in the text-

Statistical analysis description should include homogeneity of variance (e.g. with Levene or Shapiro tests) of the distribution that was perform in order to submit data to parametric tests

Information on smoking is not explain in the text (thus the scale permitted heavy or high smoking values or just smoke and non-smoking characteristics). Healthy groups should contain also smokers to control this variables.

Results

Since the groups are different in age and smoking habits could those confounding variables influence the results??? Also, since young people has fewer systemic problems and medications taken, could all of this had an influence in results?

Data of table 2 is presented in medians, which are normally associated with non-parametric statistical tests…. Why it was performed ANOVA and t tests which are parametic tests

The objective of testing consistency parameters of the tests is missing.

Discussion- phrase starting at line 344 is confusing and can be improved.

Author Response

Dear reviewer,

We appreciate your thoughtful review of our manuscript. We have carefully considered your comments and suggestions, and we have made revisions accordingly. Please find our responses to your specific points below:

The title should use only the non-abbreviated form of words.

aMMP-8 abbreviation has been deleted.

The abstract should include the meaning of the abbreviated forms of words when they are first introduced in the text (e.g. MMP, IFMA, etc). The objectives are not related to the group control and the results do not mention at all. Time of measuring in this group should be explained (only time 0  measurements) and the meaning of including this group (to test the consistency of diagnostic test? Conclusions: cannot give this conclusion just with the present study results.

The abstract has been revised.

Introduction- phrase initiating in line 56 is too long and confusing. Please divide into shorter sentences.

The phrase has been revised (Line 61-67).

Objectives should include consistency characteristics of the diagnostic tests.

The phrase has been revised (Line 105).

Material and Methods- Quantitative chairside PoC aMMP-8 analyzes were done in control patients at what end point? Please include the purpose of introducing control group

The phrase has been revised (Line 150-154).

Surgical treatment was performed in how many periodontal patients? Tests were performed 1 month after surgical treatment. Is this treatment could influence the results? Please explain better in the text-

The results section has been revised (Line 209-210).

Statistical analysis description should include homogeneity of variance (e.g. with Levene or Shapiro tests) of the distribution that was perform in order to submit data to parametric tests

Statistical analysis section has been revised (Line 189-190). We used normality tests to test the normality of the data by Shapiro–Wilk test before using parametric tests. But in our analysis both parametric and non-parametric paired samples tests were both significant by p<0.001, which provides some little bit more confidence to us to trust that our results are not dependent on which test was used.

Table 2 revised:

Pre-treatment 

Post-treatment 

Significance of comparison 

BOP%  

Mean (SD) 

76.46 (13.28) 

5.38 (6.20) 

p<0.001 

PD 

Mean (SD) 

4.33 (0.60) 

2.80 (0.21) 

p<0.001 

PI 

Mean (SD) 

1.66 (0.45) 

0.86 (0.21) 

p<0.001 

CAL  

Mean (SD) 

6.51 (1.90) 

5.32 (1.64) 

p<0.001 

aMMP-8 ng/ml 

Mean (SD) 

92.04 (72.19) 

10.0 (11.74) 

p<0.001 

aMMP-8 > 20 ng/ml 

n (%) 

23 (85,2%) 

5 (18,5%) 

Information on smoking is not explain in the text (thus the scale permitted heavy or high smoking values or just smoke and non-smoking characteristics). Healthy groups should contain also smokers to control this variables.

Information in regard to smoking has been mentioned in the text (Line 356-357).

We mentioned the absence of smokers in the healthy group when we discussed the limitations of the study. (Line 338-340)

Since the groups are different in age and smoking habits could those confounding variables influence the results??? Also, since young people has fewer systemic problems and medications taken, could all of this had an influence in results?

In this study, when selecting the healthy group, the aim was to target individuals who are young, systemically healthy, and knowledgeable dental students with an "absolute periodontal health" status and without periodontal disease experience. In this way, it was also possible to evaluate the differences between clinical periodontal parameters and chairside diagnostic test results before and after 1-month anti-infective periodontal treatment comparing with idealized systemic and periodontal health state.

On the other hand, this study is a longitudinal study that includes the first follow-up visit after anti-infective periodontal treatment. As mentioned in the text, due to various reasons (as you have also pointed out, these may include systemic diseases and various other factors) the short follow-up period did not allow for observing idealized healing in all patients. Further research is needed to investigate the effect of systemical diseases, age etc. on chairside PoC aMMP-8 tests.

Data of table 2 is presented in medians, which are normally associated with non-parametric statistical tests…. Why it was performed ANOVA and t tests which are parametic tests

We used normality tests to test the normality of the data by Shapiro–Wilk test before using parametric tests. But in our analysis both parametric and non-parametric paired samples tests were both significant by p<0.001, which provides some little bit more confidence to us to trust that our results are not dependent on which test was used.

Table 2 revised:

Pre-treatment 

Post-treatment 

Significance of comparison 

BOP%  

Mean (SD) 

76.46 (13.28) 

5.38 (6.20) 

p<0.001 

PD 

Mean (SD) 

4.33 (0.60) 

2.80 (0.21) 

p<0.001 

PI 

Mean (SD) 

1.66 (0.45) 

0.86 (0.21) 

p<0.001 

CAL  

Mean (SD) 

6.51 (1.90) 

5.32 (1.64) 

p<0.001 

aMMP-8 ng/ml 

Mean (SD) 

92.04 (72.19) 

10.0 (11.74) 

p<0.001 

aMMP-8 > 20 ng/ml 

n (%) 

23 (85,2%) 

5 (18,5%) 

The objective of testing consistency parameters of the tests is missing.

Post hoc power analysis or consistency parameters are not necessary we think as we found statistically significant results with almost all tests and thus the power and sample size were well enough. Moreover, many studies have shown that post hoc power calculations are not useful (Hoenig and Heisey 2001, Althouse 2020). The main problem is that post hoc power calculations are actually determined by the p-value. High p-values that is non-significance will always have low power while low p-values that is significant test result will always have high power. There isn’t much to learn from post hoc power calculations.

Hoenig JM, Heisey DM. The abuse of power: the pervasive fallacy of power calculations for data analysis. Am Stat. 2001; 55:19-24.

Althouse A. Post Hoc Power: Not Empowering, Just Misleading. J Surg Res. 2021 Mar; 259:A3-A6

Discussion- phrase starting at line 344 is confusing and can be improved.

The phrase has been revised (Line 379-382).
